# Lyapunov Drift-Plus-Penalty-Based Cooperative Uplink Scheduling in Dense Wi-Fi Networks

**DOI:** 10.3390/s24082399

**Published:** 2024-04-09

**Authors:** Yonggang Kim, Yohan Kim

**Affiliations:** 1Division of Computer Science and Engineering, Kongju National University, Cheonan 31080, Republic of Korea; ygkim@kongju.ac.kr; 2Department of Software, Dongseo University, Busan 47011, Republic of Korea

**Keywords:** cooperative uplink scheduling, Lyapunov drift-plus-penalty, trade-off modeling, Wi-Fi networks

## Abstract

In high-density network environments with multiple access points (APs) and stations, individual uplink scheduling by each AP can severely interfere with the uplink transmissions of neighboring APs and their associated stations. In congested areas where concurrent uplink transmissions may lead to significant interference, it would be beneficial to deploy a cooperative scheduler or a central coordinating entity responsible for orchestrating cooperative uplink scheduling by assigning several neighboring APs to support the uplink transmission of a single station within a proximate service area to alleviate the excessive interference. Cooperative uplink scheduling facilitated by cooperative information sharing and management is poised to improve the likelihood of successful uplink transmissions in areas with a high concentration of APs and stations. Nonetheless, it is crucial to account for the queue stability of the stations and the potential delays arising from information exchange and the decision-making process in uplink scheduling to maintain the overall effectiveness of the cooperative approach. In this paper, we propose a Lyapunov drift-plus-penalty framework-based cooperative uplink scheduling method for densely populated Wi-Fi networks. The cooperative scheduler aggregates information, such as signal-to-interference-plus-noise ratio (SINR) and queue status. During the aggregation procedure, propagation delays are also estimated and utilized as a value of expected cooperation delays in scheduling decisions. Upon aggregating the information, the cooperative scheduler calculates the Lyapunov drift-plus-penalty value, incorporating a predefined model parameter to adjust the system accordingly. Among the possible scheduling candidates, the proposed method proceeds to make uplink decisions that aim to reduce the upper bound of the Lyapunov drift-plus-penalty value, thereby improving the network performance and stability without a severe increase in cooperation delay in highly congested areas. Through comprehensive performance evaluations, the proposed method effectively enhances network performance with an appropriate model parameter. The performance improvement is particularly notable in highly congested areas and is achieved without a severe increase in cooperation delays.

## 1. Introduction

To facilitate wireless connectivity for devices such as mobile phones, vehicles, or IoT devices, Wi-Fi networks are established through the deployment of access points (APs). These APs independently handle uplink transmissions, collecting information from associated stations within their coverage area and scheduling the uplink transmissions accordingly. In densely populated areas where multiple APs and stations are distributed, the independent operation of multiple APs can significantly degrade the network performance due to interference originating from nearby communications. One strategy to improve per-node throughput in such congested environments is to leverage alternative wireless networks, such as cellular networks, to offload transmission requests [1]. However, integrating and efficiently utilizing heterogeneous network architectures can present deployment challenges due to the complexity of managing multiple network types. Without relying on other network infrastructures, an alternative approach involves the joint design and adjustment of the transmission power of Wi-Fi-enabled nodes to enhance throughput [2]. Additionally, transmission requests could be adjusted to balance the load among APs for improved stability and throughput [3]. However, achieving efficient and enhanced network performance becomes challenging when there is a lack of shared information on channel states or scheduling decisions among the APs and stations. In such scenarios, uplink transmissions from stations are particularly vulnerable to being significantly degraded by interference from nearby communications between other APs and stations. To alleviate the performance degradation issue in such congested areas, a cooperative scheduler or a central coordinating entity could be deployed to aggregate the uplink information and coordinate uplink scheduling. By centralizing the functions, such as baseband signal processing and uplink scheduling, more efficient network-wise scheduling decisions can be made. This centralized or cooperative approach is anticipated to outperform traditional non-cooperative Wi-Fi networks, especially in areas with intense uplink requests. For instance, rather than processing all uplink requests independently as performed in non-cooperative Wi-Fi systems, the cooperative scheduler could selectively manage the numerous uplink requests in areas where network coverage overlaps among densely deployed proximate APs. Moreover, to enhance the signal-to-interference-plus-noise ratio (SINR), multiple neighboring APs could be allocated to a single station, utilizing diversity combining techniques to improve the quality of the received signal. This cooperative approach to network management is likely to be more effective than independent network management by individual APs, particularly in densely populated network environments with numerous APs and stations.

Previous research has delved into medium access control (MAC) protocols that support diversity combining for uplink transmissions by the collaborative efforts of multiple APs [4]. In vulnerable wireless channel environments, a central unit could allocate several APs in proximate to support the uplink transmission from a single station. The signals received by these cooperative APs are then combined to improve SINR, facilitating successful decoding. Such cooperative uplink scheduling has been shown to potentially yield better SINR and data rates for uplink transmissions. However, previous research focused on many-to-one scenarios where multiple APs support a single station without considering the presence of other APs and stations. This paper investigates cooperative uplink scheduling within the context of dense Wi-Fi networks with multiple APs and stations. In such networks, the aim is to enhance both the success probability of individual uplink transmissions and the provision for the high volume of uplink requests from multiple stations. In scenarios where a cooperative scheduler is employed, the cooperative uplink scheduling is performed in consideration of the anticipated success probability of each feasible uplink transmission candidate and the corresponding network-wise stability. Note that additional delays for aggregating information and disseminating scheduling decisions are required in cooperative uplink scheduling. The proposed method employs a Lyapunov drift-plus-penalty framework to model the network-wide stability of uplink transmissions across multiple stations and the delay inherent in cooperative scheduling.

The remainder of this paper is organized as follows. Section 2 reviews related work on uplink transmission methods in Wi-Fi networks. Section 3 describes the system model for cooperative uplink transmission scenarios. Section 4 details the proposed Lyapunov drift-plus-penalty-based cooperative uplink scheduling method. Section 5 presents an evaluation of performance, and Section 6 concludes the paper.

## 2. Related Work

The increasing demand for wireless connectivity services in densely populated areas, where numerous APs and stations are present, necessitates efficient network management to enhance user experience and throughput performance. Kosek-Szott and Domino investigated the issue of performance degradation due to excessive contention in dense Wi-Fi networks [5]. The authors proposed a novel back-off mechanism that adjusts window size based on the ratio of unsuccessful and unallocated resource units, thereby reducing collision likelihood. However, the independent operation of APs may prove inadequate due to heightened channel uncertainty caused by a growing number of independently operating nodes. Thus, cooperative or centralized strategies have garnered significant research interest because they have the potential to enhance wireless service performance in densely populated networks. Focusing on implementing cooperative management, Zhang et al. underscore the benefits of collaborative resource management within a software-defined Wi-Fi network architecture in [6]. The authors demonstrated the feasibility of establishing a software-defined architecture that decouples the control and data planes for efficient network management within densely populated areas. By deploying global and local controllers for collaborative resource management, the authors showed that the well-designed control mechanism of coexistence between centralized and distributed network management could improve the throughput performance. Previous studies have presented the issues of providing wireless connectivity services in densely populated areas, suggesting that cooperative or centralized management could significantly improve Wi-Fi services in such environments with practical implementation. Based on the findings in these studies, we focus on cooperative management and related considerations regarding transmission success probability, stability, and required delay for cooperation.

The cooperative resource management facilitated by information sharing among deployed nodes has the potential to efficiently improve the throughput performance of Wi-Fi services in densely populated areas. Shen et al. explored queue-aware resource allocation in centralized Wi-Fi networks, where a primary AP manages other APs, aiming to maximize the aggregate rate of uplink and downlink transmissions [7]. The proposed method prioritizes stations with longer queue lengths while concurrently reducing collision risks. Furthermore, with the advent of learning technologies, learning-based resource management has been investigated in Wi-Fi networks. Zhang et al. delved into deep reinforcement learning for AP coordination in dense overlapping Wi-Fi networks, advocating for a centralized AP controller that employs deep reinforcement learning to refine back-off decisions and mitigate collision frequency [8]. Wydmański and Szott presented deep reinforcement learning approaches for contention window optimization, introducing two centralized control methods based on deep Q network (DQN) and deep deterministic policy gradient (DDPG) to maintain stable throughput as the number of station surges [9]. The issue of channel assignment in dense Wi-Fi networks was also investigated. Kinoshita et al. addressed the learning-based channel assignment system managed by a central broker to enhance user throughput and fairness across mobile and Wi-Fi networks [10]. The broker employs a genetic algorithm for channel allocation between the two network types in the considered heterogeneous networks. Sangdeh and Zeng investigated deep learning for channel sounding and downlink resource allocation [11]. The proposed method allocates resource units and transmission power to improve downlink performance by utilizing channel reciprocity and limited state information to train models. On the other hand, the proposed method takes into account the trade-off relationships in modeling the scheduling problem. As more APs are incorporated into cooperative management, performance enhancement of transmission stability and the probability of successful transmission for each scheduled transmission could be achieved with the increased diversity in scheduling decisions. However, the increase in cooperation also brings about a rise in delays due to the time required for information aggregation and decision broadcasting. Note that such an increase in delay may impede the feasibility of cooperative Wi-Fi systems in real-world environments. To design the scheduling decisions with consideration of transmission stability and cooperation delay, we employ a Lyapunov drift-plus-penalty-based framework, which efficiently models the balancing problem between two metrics with respect to the queue changes of the nodes. Furthermore, the proposed Lyapunov drift-plus-penalty-based method has the potential to be combined with other methods from previous studies in problem formulation and solving.

The cooperative scheduling approach has also been investigated for various applications and scenarios beyond Wi-Fi networks. Mwakwata et al. investigated a cooperative scheduler-based interference avoidance method in NB-IoT systems [12]. In the considered scenario, cellular base stations supporting NB-IoT services share their scheduling table for inter-cell interference avoidance scheduling. Mwakwata et al. also investigated the impact of a designed cooperative scheduler for orthogonal multiple access and non-orthogonal multiple access scenarios in cellular networks with massive connectivity [13]. Li and Cai investigated cooperative control methods for uplink transmissions in cellular networks by considering device-to-device relay transmissions and network coding in dense areas [14]. Based on the stochastic geometry modeling and density, multiple user devices are cooperatively and randomly utilized for relaying for uplink transmissions, or network coding is further adopted. Hua et al. investigated cooperative scheduling for pilot reuse among neighboring cells in massive MIMO systems [15]. The proposed method cooperatively schedules users to share the limited orthogonal pilots among neighboring cells. As noted in related studies, cooperative scheduling has been shown to effectively enhance network performance, particularly in congested or densely populated areas. Although Wi-Fi networks have distinct characteristics compared to cellular networks, such as differences in backhaul connections, deploying a cooperative scheduler or a central coordinating entity for cooperative scheduling could lead to improved Wi-Fi performance.

Cooperative resource management indeed shows promise for enhancing connectivity services in dense Wi-Fi networks, where the coordination of resources can lead to improved uplink throughput and overall network performance. However, the introduction of cooperative strategies often brings about additional delays due to the need for information sharing and decision-making among APs. This can counteract some of the benefits gained from cooperation. In an effort to address these delays, the previous study has explored MAC protocols for cooperative uplink transmissions that accommodate delay constraints by preemptively sending immediate ACK manages based on the anticipated uplink success probability [4]. The designed MAC protocol transmits ACK messages before completing the data reception under the specific criteria to reduce delays required for cooperative transmissions. However, while these solutions may work well in scenarios where multiple APs assist a single station, there is a gap in addressing network-wide scheduling decisions in more complex many-to-many scenarios. These involve densely deployed APs and stations where multiple APs may have to cooperate with multiple stations.

In this paper, we propose a new method for cooperative uplink scheduling that leverages a Lyapunov drift-plus-penalty framework, specifically designed for dense Wi-Fi networks. Lyapunov drift-plus-penalty-based optimization is a well-known method used to solve problems that are formulated based on trade-off relationships in stochastic systems [16]. The problem is formulated to minimize the upper bound of the objective function based on the stochastic change model for the considered objectives, such as stability or utility, and the trade-off parameter between the objectives. We utilize the Lyapunov drift-plus-penalty framework and take into consideration both the probability of successful uplink transmissions and the expected delays from cooperative scheduling. With this approach, we aim to enable efficient scheduling decisions that can handle the complexity of many-to-many interactions between APs and stations. By applying the Lyapunov drift-plus-penalty-based framework, we strive to strike a balance between enhancing uplink throughput performance and managing the incremental delays introduced by the necessity for cooperation. The framework helps to ensure that the network remains stable and that queues at stations do not grow indefinitely while alleviating the time required for information aggregation and scheduling decisions. Through the proposed Lyapunov drift-plus-penalty-based framework, we achieve an effective balance between improved uplink throughput performance and the incremental delays necessitated by cooperation.

## 3. System Model

### 3.1. Network Topology and Non-Cooperative Uplink Transmissions

In the considered scenario of uplink transmissions within dense Wi-Fi networks, the system architecture comprises a cooperative scheduler, multiple APs, and various non-AP stations, as depicted in Figure 1. The cooperative scheduler, denoted as *c*, orchestrates the uplink scheduling of APs in $A=\{a_{1},a_{2},\cdots ,a_{A}\}$ to cooperatively facilitate wireless connectivity services for stations in $S=\{s_{1},s_{2},\cdots ,s_{S}\}$ in the network.

In centralized Wi-Fi network architecture, densely deployed stations transmit uplink data to APs through wireless channels, and the APs may forward received data to a cooperative scheduler through wired fronthaul networks that connect the cooperative scheduler and APs. Let $x_{s_{l}}$ and $p_{s_{l}}$ be the uplink data and transmit power from station $s_{l}$, respectively. Then, the receive signal at the AP $a_{i}\in A$ for the single channel scenario is denoted as follows:(1)$$y_{s_{l},a_{i}}=h_{s_{l},a_{i}}\sqrt{p_{s_{l}}}x_{s_{l}}+\sum_{s_{l^{\prime }}\in S/s_{l}}h_{s_{l^{\prime }},a_{i}}\sqrt{p_{s_{l^{\prime }}}}+z_{a_{i}}$$
where $h_{s_{l},a_{i}}$ is the channel coefficient from station $s_{l}$ to AP $a_{i}$ and $z_{a_{i}}$ is the additive white Gaussian noise with variance $\sigma^{2}$ at the AP $a_{i}$. If we assume that a single AP is associated with a single station to provide wireless connectivity services for uplink transmissions, then the SINR of the received signal (Equation 1) is represented as follows:(2)$$\gamma_{s_{l},a_{i}}=\frac{p_{s_{l}}{|h_{s_{l},a_{i}}|}^{2}}{\sum_{s_{l^{\prime }}\in S\setminus s_{l}}p_{s_{l^{\prime }}}{|h_{s_{l^{\prime }},a_{i}}|}^{2}+\sigma^{2}}.$$
When the SINR $\gamma_{s_{l},a_{i}}$ of the uplink signal from $s_{l}\in S$ is greater than SINR threshold $\gamma_{th}$, the received signal is highly expected to be successfully decoded.

To improve success probability of uplink transmissions of the uplink signal $y_{s_{l},a_{i}}$ with SINR $\gamma_{s_{l},a_{i}}$, we can increase numerator in (Equation 2) while decreasing denominator in (Equation 2) in dense centralized Wi-Fi networks. Multiple APs can cooperatively support the uplink transmission from the same station by receiving the uplink signal simultaneously and forwarding the received signal to the cooperative scheduler. With the aggregated uplink data originating from the same station, the cooperative scheduler may perform diversity combining techniques to improve SINR and decode the received signal with the improved SINR or apply majority voting to correctly infer the transmitted data from the noise-added data [4,17]. Furthermore, the cooperative scheduler may select a subset of stations for uplink transmissions instead of scheduling all stations simultaneously. By strategically scheduling specific stations, the interference among uplink transmissions can be reduced in dense networks.

### 3.2. Delay-Constrained Cooperative Transmissions

In dense centralized Wi-Fi networks, information such as queue status or channel states is collected and forwarded to the cooperative scheduler for cooperative uplink transmissions. The aggregation of information at the cooperative scheduler enables intelligent scheduling decisions for handling highly congested uplink transmission requests in the networks. Let $B_{s_{l}}\left(t\right)$ be the uplink data queue of station $s_{l}\in S$ at time *t*. $B_{s_{l}}\left(t\right)$ and $h_{s_{l},a_{i}}$ for $s_{l}\in S$ and $a_{i}\in A$ are transferred to the cooperative scheduler for cooperative uplink transmissions.

During the data aggregation, the propagation delays from stations to the cooperative scheduler are also estimated. Both the propagation delays in wireless channels and wired fronthaul networks may affect the performance of wireless connectivity services. When the propagation delay is longer than a certain threshold, stations do not receive ACK messages within the time threshold even though the uplink data are successfully decoded. Note that although the data originated from the same station, uplink data are received by multiple cooperative APs and may experience different wireless channels and routing paths in wired fronthaul networks.

Let the propagation delay from station $s_{l}$ and AP $a_{i}$ to the cooperative scheduler *c* in the wired fronthaul network be $d_{s_{l},a_{i},c}$. Because data from multiple paths are cooperatively combined within a certain time threshold $d_{th}$ for improved SINR, APs with feasible propagation delays should be selected to support the same station. Among the feasible APs satisfying the $d_{s_{l},a_{i},c}<d_{th}$, the cooperative scheduler determines the cooperative AP set for supporting uplink transmission from station $s_{l}$ so that the maximum delay is shorter than or $d_{th}$. If we denote the selected AP set for cooperatively supporting station $s_{l}$ as ${\hat{A}}_{s_{l}}$, then ${\hat{A}}_{s_{l}}$ could be defined as follows:(3)$${\hat{A}}_{s_{l}}=\left\{a_{i}\;|\;d_{s_{l},a_{i},c}<d_{th},\;a_{i}\in A\right\}.$$
Since we perform cooperative transmission for each transmission channel, ${\hat{A}}_{s_{l}}$ satisfies $\{{\hat{A}}_{s_{1}},{\hat{A}}_{s_{2}},\cdots ,{\hat{A}}_{s_{S}}\}\subseteq A$ and ${\hat{A}}_{s_{l}}\cap {\hat{A}}_{s_{m}}=\emptyset$ for $s_{l},s_{m}\in S$ in the considered scenario. Then, the aggregated SINR at the cooperative scheduler, *c*, performing diversity combining becomes
(4)$$\gamma_{s_{l},{\hat{A}}_{s_{l}}}=\frac{\sum_{a_{i}\in {\hat{A}}_{s_{l}}}p_{s_{l}}{|h_{s_{l},a_{i}}|}^{2}}{\sum_{a_{i}\in {\hat{A}}_{s_{l}}}\sum_{s_{l^{\prime }}\in \hat{S}\setminus s_{l}}p_{s_{l^{\prime }}}{|h_{s_{l^{\prime }},a_{i}}|}^{2}+\sigma^{2}}.$$
When the gain of the desired received signal strength is higher than the increase in interference, then the SINR satisfies $\gamma_{s_{l},{\hat{A}}_{s_{l}}}\geq \gamma_{s_{l},a_{i}}$. In dense networks, when multiple access points receive signals from station $s_{l}$ for diversity combining at the cooperative scheduler while a smaller number of stations are scheduled appropriately, the likelihood of successful decoding of scheduled uplink transmissions would increase compared to a non-cooperative system. Based on the expected uplink success probability, the uplink scheduling decision could be made to improve network-wise transmission stability.

The signals received at APs in ${\hat{A}}_{s_{l}}$ are forwarded and combined at the cooperative scheduler *c* for decoding the uplink signals from the station $s_{l}$. Although the cooperative AP set for $s_{l}inS$, ${\hat{A}}_{s_{l}}$, is defined not to avoid delay constraint $d_{th}$, the required delay is better to be decreased for efficient cooperative scheduling. We denote the maximum transmission delay for providing cooperative wireless connectivity services to the station $s_{l}$ as $d_{s_{l}}^{max}$. Then, $d_{s_{l}}^{max}$ is represented as follows:(5)$$d_{s_{l}}^{max}=\max\left\{d_{s_{l},a_{i},c}\right\}\;\;\mathrm{for}\;\;a_{i}\in {\hat{A}}_{s_{l}}.$$
The proposed method performs uplink scheduling with consideration of both uplink transmission success probability affected by SINR and the required delay for uplink cooperation.

## 4. Uplink Transmissions in Cooperative Networks

### 4.1. Uplink Transmission Procedure

In dense networks where stations and APs are closely deployed, independent uplink transmissions from stations to their associated APs can cause interference and degrade network performance. To mitigate the interference among uplink transmission in proximate areas, instead of scheduling all uplink transmissions simultaneously, a cooperative scheduler could be established to select a subset of stations for uplink transmissions and coordinate APs to cooperatively support the selected stations. The process of uplink transmissions in the considered cooperative Wi-Fi networks, with a cooperative scheduler managing operations of APs in A and stations in set S, is illustrated in Figure 2. After aggregating the information on queue status and channel conditions, the cooperative scheduler performs Lyapunov drift-plus-penalty-based cooperative uplink scheduling.

A.The cooperative scheduler may decide to perform cooperative scheduling in scenarios where a significant amount of interference between stations and APs is anticipated to considerably degrade network throughput performance. To ensure efficient wireless connectivity service provision with appropriate cooperative uplink scheduling, queue status, and channel state information are consolidated before scheduling decisions are made. Prior to managing uplink transmissions in the networks, the cooperative scheduler requests stations to send status reports to gather information on queue status, channel state, and propagation delay through the request-to-send and clear-to-send processes. Note that status report messages are transmitted to the cooperative scheduler through both wireless and wired channels, allowing for the estimation of propagation delay from the station to the cooperative scheduler. It is assumed that the propagation delay remains symmetric during the time interval between forwarding status reports and actual uplink transmissions from the scheduled stations. In Figure 2, five stations from $s_{1}$ to $s_{5}$ transmit status report messages to the cooperative scheduler through deployed APs A. By aggregating status report messages, the cooperative scheduler gathers information on queue status $B_{s_{l}}\left(t\right)$, channel state $h_{s_{l},a_{i}}$, and propagation delay $d_{s_{l},a_{i},c}$ for $s_{l}\in S$ and $a_{i}\in A$.B.After aggregating information such as queue status, channel states, and propagation delay, the cooperative scheduler chooses a subset of stations for uplink transmissions and designates multiple APs for cooperative uplink support. Rather than selecting all stations attempting to transmit uplink data, the cooperative scheduler may opt for a subset of stations to mitigate interference among uplink transmissions in dense networks. Moreover, assigning multiple APs in close proximity to the selected station improves SINR and increases the likelihood of successful decoding of the transmitted data. As shown in Figure 2, the cooperative scheduler selects a subset of stations and assigns cooperative APs, which are represented as $\left\{{\hat{A}}_{s_{1}},{\hat{A}}_{s_{2}},\cdots ,{\hat{A}}_{s_{5}}\right\}$. The APs in ${\hat{A}}_{s_{l}}$ cooperatively support uplink transmission from station $s_{l}$ for $s_{l}\in \hat{S}$. Note that ${\hat{A}}_{s_{l}}=\emptyset$ for $s_{l}\notin \hat{S}$ where $\hat{S}$ represents a set of stations allowed to transmit uplink data.C.The cooperative scheduler notifies the selected stations of the uplink scheduling decision by transmitting triggering messages. The selected stations transmit uplink data in their queues to APs through wireless channels. In the proposed uplink transmission method, uplink transmission is decided based on Lyapunov drift-plus-penalty modeling that considers queue changes, uplink success probabilities, and delays. In Figure 2, stations $s_{1}$, $s_{2}$, and $s_{5}$ transmit uplink data while stations $s_{3}$ and $s_{4}$ are not allowed for uplink transmissions. Note that the proposed drift-plus-penalty-based approach calculates the expected uplink success probability and corresponding queue changes to provide stable wireless connectivity services with high throughput performance. The performance of the queuing system and the stability of the Lyapunov drift-plus-penalty approach are analyzed in [16]. In the proposed algorithm, the proposed method selects the solution with consideration of queue arrival and processing rates for throughput performance and network-wise stability. Because the stations are scheduled to increase both throughput performance and stability by avoiding queue overflow, the stations with large queue lengths are highly likely to be selected for uplink transmissions.D.The cooperative APs forward the received data to the cooperative scheduler through wired fronthaul networks. Note that uplink data originating from the same station are forwarded through multiple paths in fronthaul networks. After the uplink data decoding is completed, the ACK messages are transmitted to the stations. In Figure 2, cooperative APs in ${\hat{A}}_{s_{1}}$, ${\hat{A}}_{s_{2}}$, and ${\hat{A}}_{s_{5}}$ receives uplink data from station $s_{1}$, $s_{2}$, and $s_{5}$, respectively. The APs forward the received data to the cooperative scheduler for diversity combining so that the success probability of decoding the uplink data increases with the improved SINR.

In a dense network where stations and APs are densely deployed, there is massive interference among stations and APs. For uplink transmission scenarios, independent operations of APs may degrade the success probability of uplink transmissions in a network. Instead of independent operations for providing wireless connectivity services to stations, the cooperative scheduler may centrally manage the uplink operations of APs. Because the distributions of propagation delays affect the Wi-Fi service performances, the proposed method considers both the propagation delays and success probability for uplink scheduling decisions.

### 4.2. Lyapunov Drift-Plus-Penalty-Based Uplink Modeling and Decision

In this study, we consider an uplink transmission scenario where densely deployed stations try to transmit uplink data. After the stations transmit uplink data in their queues, the queue status of each station changes depending on the queue arrivals and transmission success probability. Let $B_{s_{l}}\left(t\right)$ denote the uplink data queue of station $s_{l}\in S$ at time *t* as described in Section 3. Then, the queue status at time t+1 is expressed as follows:(6)$$B_{s_{l}}\left(t+1\right)=\max\left\{B_{s_{l}}\left(t\right)+\theta_{s_{l}}\left(t\right),0\right\},\;\;\forall s_{l}\in S$$
where θ(t) is the queue change function at time *t*. The expectation of queue change function $\theta_{s_{l}}\left(t\right)$ is as follows:(7)$$E\left\{\theta_{s_{l}}\left(t\right)\right\}=E\left\{\alpha_{s_{l}}\left(t\right)-P_{u_{l}}^{succ}\beta_{s_{l}}\left(t\right)\right\},\;\;\forall s_{l}\in S$$
where $\alpha_{s_{l}}\left(t\right)$ and $\beta_{s_{l}}\left(t\right)$ are uplink data arrivals and processing rates, respectively. In (Equation 7), $P_{u_{l}}^{succ}$ denotes the transmission success probability of uplink data. The transmission success probability $P_{u_{l}}^{succ}$ depends on the SINR $\gamma_{s_{l},{\hat{A}}_{s_{l}}}$ and the SINR threshold $\gamma_{th}$. Let $P_{u_{l}}^{out}$ be the outage probability where $P_{s_{l}}^{succ}=1-P_{s_{l}}^{out}$. Then, the outage probability of uplink transmission from station $s_{l}$ is expressed as
(8)$$\begin{array}{cc} P_{s_{l}}^{out} & =P(\gamma_{s_{l},{\hat{A}}_{s_{l}}}<\gamma_{th})\\ & =P\left(\frac{\sum_{a_{i}\in {\hat{A}}_{s_{l}}}p_{s_{l}}{|h_{s_{l},a_{i}}|}^{2}}{\sum_{a_{i}\in {\hat{A}}_{s_{l}}}\sum_{s_{l^{\prime }}\in \hat{S}\setminus s_{l}}p_{s_{l^{\prime }}}{|h_{s_{l^{\prime }},a_{i}}|}^{2}+\sigma^{2}}<\gamma_{th}\right)\\ & =1-P\left(\frac{\sum_{a_{i}\in {\hat{A}}_{s_{l}}}p_{s_{l}}{|h_{s_{l},a_{i}}|}^{2}}{\gamma_{th}}\geq \sum_{a_{i}\in {\hat{A}}_{s_{l}}}\sum_{s_{l^{\prime }}\in \hat{S}\setminus s_{l}}p_{s_{l^{\prime }}}{|h_{s_{l^{\prime }},a_{i}}|}^{2}+\sigma^{2}\right). \end{array}$$
Romero-Jerez et al. derive the outage probability of maximal ratio combining in Nakagami-m fading environments and also describe the closed form of the outage probability when the fading parameter of Nakagami-m is equal to 1 in [18]. The signals experience a Rayleigh fading channel when the Nakagami fading parameter *m* is 1, and the outage probability in (Equation 8) becomes
(9)$$\begin{array}{cc} P_{s_{l}}^{out}= & 1-e^{-\frac{\sigma^{2}}{{\hat{p}}_{0}}}\sum_{k=0}^{|{\hat{A}}_{s_{l}}|}{\left(\frac{1}{{\hat{p}}_{0}}\right)}^{k}\times \sum_{\sum_{i=0}^{|A\setminus {\hat{A}}_{s_{l}}|}k_{i}=k}\frac{2k_{0}}{k_{0}!}\prod_{i=1}^{|A\setminus {\hat{A}}_{s_{l}}|}\frac{1}{{\hat{p}}_{i}}{\left[\frac{1}{{\hat{p}}_{0}}+\frac{1}{{\hat{p}}_{i}}\right]}^{-k_{i}-1} \end{array}$$
where ${\hat{p}}_{0}$ and ${\hat{p}}_{i}$ are the received desired signal power and interference power normalized by $\gamma_{th}$. Note that m=1 if the Rayleigh fading channel is assumed. The outage probability $P_{s_{l}}^{out}$ is a submodular function of which outage probability decreases as SINR increases. With the expected queue change in (Equation 7) followed by $P_{s_{l}}^{succ}=1-P_{s_{l}}^{out}$, the proposed method decides cooperative uplink transmissions in dense centralized Wi-Fi networks.

In the considered cooperative uplink transmission scenarios, multiple stations try to transmit uplink in their queues while the deployed APs cooperatively provide wireless connectivity services to the selected subset of stations. Because multiple cooperative APs could receive uplink signals, path diversity increases, resulting in enhanced uplink transmissions’ robustness with improved SINR. The expected amount of data with successful uplink transmissions is obtained by calculating the expected difference in queue status from two consecutive time slots. The expected queue status is calculated with the current queue status and the expected successful probability of uplink transmissions as in Equation 7. However, the increase in path diversity may also increase the required delay for diversity combining at the cooperative scheduler because data forwarding is required between APs and the cooperative scheduler. The proposed method models the trade-off relations between the successful uplink transmissions and the required delay by utilizing the Lyapunov-drift-plus-penalty approach. We define the Lyapunov function L(t) as a measure of the total queue backlog, where
(10)$$L\left(t\right)=\frac{1}{2}\sum_{s_{l}\in S}{B_{s_{l}}\left(t\right)}^{2}.$$
Using (Equation 10), the Lyapunov drift ΔL(t) is defined as L(t+1)−L(t). Then, Lyapunov drift ΔL(t) is expressed as follows:(11)$$\begin{array}{cc} \Delta L(t) & =L(t+1)-L(t)\\ & \leq \frac{1}{2}\sum_{s_{l}\in S}{\theta_{s_{l}}\left(t\right)}^{2}+\sum_{s_{l}\in S}B_{s_{l}}\left(t\right)\theta_{s_{l}}\left(t\right)\\ & \leq U+\sum_{s_{l}\in S}B_{s_{l}}\left(t\right)\theta_{s_{l}}\left(t\right) \end{array}$$
where *U* is the upper bound of $\frac{1}{2}\sum_{s_{l}\in S}{\theta_{s_{l}}\left(t\right)}^{2}$. Note that the upper bound of ${\theta_{s_{l}}\left(t\right)}^{2}$ for station $s_{l}\in S$ is affected by the queue arrival rate, and we assume that the maximum queue arrival rate does not exceed the maximum buffer size. The inequality in (Equation 11) contains information about the queue change of stations and success probabilities of uplink cooperative transmissions followed by uplink scheduling decisions.

For diversity combining, the same data originating from the same station could be forwarded to the cooperative scheduler through multiple routing paths where each path includes an AP participating in cooperative uplink transmissions. In Section 3, the penalty of required delay for cooperative uplink scheduling is modeled as $d_{s_{l}}^{max}$ for $s_{l}\in S$. We add the penalty term to (Equation 11); then, the Lyapunov drift-plus-penalty is expressed as follows:(12)$$\Delta L\left(t\right)+V\sum_{s_{l}\in S}d_{s_{l}}^{max}\leq U+V\sum_{s_{l}\in S}d_{s_{l}}^{max}+\sum_{s_{l}\in S}B_{s_{l}}\left(t\right)\theta_{s_{l}}\left(t\right)$$
where *V* is a non-negative weight that adjusts the relation between the Lyapunov drift term and the penalty term. The proposed uplink scheduling method minimizes the upper bound of the Lyapunov drift-plus-penalty denoted in (Equation 12) to stably achieve enhanced network throughput without a significant increase in required delay for cooperative scheduling.

Algorithm 1 describes the simple heuristic algorithm for the Lyapunov drift-plus-penalty-based cooperative uplink scheduling decision. Because calculating all the feasible candidates for uplink scheduling may cause high computational complexity in dense networks, the simple heuristic algorithm that compares the $N_{max}$ randomly selected solutions instead of inspecting all the feasible solutions is adopted for performance evaluation. Let κ represent the number of stations scheduled for the current time slot. Given that a subset of stations, i.e., κ≤S, is chosen for uplink transmissions, there are C(S,κ) possible candidates for selecting stations, where C denotes the combination calculation. Because a single channel scenario is considered in the system model, the number of cases that *A* APs are assigned to κ+1 nodes, including non-association, can be represented by the Stirling number of the second kind, denoted as S(A,κ+1). Consequently, for each κ, where κ∈{1,…,S}, there are C(S,κ)S(A,κ+1) possible scheduling decisions. As the station or AP numbers grow, the quantity of viable solutions surges drastically. To validate the advantages of using Lyapunov drift-plus-penalty-based modeling for cooperative scheduling in densely populated networks, we simply employed a randomness-based selection algorithm to demonstrate feasibility. Algorithm 1 describes a randomness-based scheduling method that can be performed in a short time in dense networks. Queue status and channel state information are aggregated during the request-to-send and clear-to-send processes described in Section 4.1. After aggregating the data, the Lyapunov drift-plus-penalty value is calculated for feasible solutions. The proposed algorithm selects the candidate with the minimum Lyapunov drift-plus-penalty value among the feasible solutions. Because Lyapunov drift calculates the expected differences in queue status between two consecutive time slots, minimizing the upper bound of Lyapunov drift-plus-penalty increases the overall network stability by avoiding significant unfairness in station selection while mitigating the delay requirements for cooperation.
**Algorithm 1** Simple heuristic algorithm for Lyapunov drift-plus-penalty-based cooperative uplink scheduling 1:Initialize the trade-off parameter *V* and the maximum iteration number $N_{max}$. 2:Aggregate queue status and channel state information. 3:**for** $i=1,2,\cdots ,N_{max}$ **do** 4:      // Lyapunov drift-plus-penalty calculation 5:      Randomly select a subset of stations and associated APs. 6:      Calculate $P_{s_{l}}^{out}=P\left(\gamma_{s_{l},{\hat{A}}_{s_{l}}}<\gamma_{th}\right)$ in (Equation 8). 7:      Calculate $E\left\{\theta_{s_{l}}\left(t\right)\right\}=E\left\{\alpha_{s_{l}}\left(t\right)-P_{u_{l}}^{succ}\beta_{s_{l}}\left(t\right)\right\},\;\;\forall s_{l}\in S$ in (Equation 7). 8:      Calculate the expected delay $d_{s_{l}}^{max}$ for $s_{l}\in S$ for cooperative scheduling 9:      Store the Lyapunov drift-plus-penalty value by calculating             $U+V\sum_{s_{l}\in S}d_{s_{l}}^{max}+\sum_{s_{l}\in S}B_{s_{l}}\left(t\right)\theta_{s_{l}}\left(t\right)$.10:      Store the scheduling decision.11:**end for**12:Select the solution with the minimum Lyapunov drift-plus-penalty value.

## 5. Performance Evaluation

We conduct performance evaluations using MATLAB-based simulations to compare the throughput and delay performance in IEEE 802.11ax-compliant networks [19]. The simulation was implemented using the WLAN system toolbox of MATLAB, which provides IEEE 802.11ax channel configurations. In a square area of 50 m on the side, 20 APs and 10 to 30 clients are randomly deployed in the network. Note that a single wireless channel without channel bonding or channel hopping is considered in the simulation scenario for simplicity. The simulation parameters are listed in Table 1. The carrier frequency and channel bandwidth are specified at 5.25 GHz and 40 MHz, respectively. A low-density parity-check code (LDPC) is adopted, with dual antennas at both the transmitter and receiver nodes. The path loss exponent is determined to be 2 for distances less than or equal to 5 m between the transmit-receive nodes and set to 3.5 for distances longer than 5 m. Rayleigh fading is employed to model the small-scale fading effects. The wireless signal transmissions are performed with a power of 25 dBm, the modulation and coding scheme (MCS) at 3, and the physical layer convergence procedure (PLCP) service data unit (PSDU) size of 1024 bytes. The background noise level is established at −80 dBm. Given that each transmitter transmits wireless signals at a power level of 25 dBm, the interference from nearby signals significantly outweighs the impact of the background noise. The propagation delays between nodes in the considered cooperative networks follow the gamma distribution with shape parameter $g_{sh}$ and scale parameter $g_{sc}$ [4]. The mean and variance of delay are calculated as $g_{sh}g_{sc}$ and $g_{sh}{g_{sc}}^{2}$, respectively. We set the shape parameter $g_{sh}$ as 4 and the scale parameter $g_{sc}$ as 2. Hence, the mean and variance of delays required for cooperative uplink transmission are 8 μs and 16 μs, respectively. In the established network topology, we compare the proposed cooperative scheduling method with other uplink scheduling methods, such as the random and non-cooperative methods. The random method randomly decides the cooperation among APs. Except for the AP associated with the station trying to transmit uplink data, the cooperative scheduler randomly selects the additional APs for uplink cooperation. The scheduling decision of the random method is based on the expected throughput without consideration of stability or cooperation delay. For fair performance comparison with the proposed Lyapunov drift-plus-penalty-based cooperative uplink scheduling method described in Algorithm 1, the random method selects the best solution among the feasible candidates obtained after $N_{max}$ random trials. On the other hand, for the non-cooperative method, APs independently support wireless connectivity service to the associated stations without cooperation. For various trade-off parameters, *V*, the proposed method is compared to other methods.

Figure 3 shows the packet error rate with regard to signal-to-noise ratio (SNR) in the constructed simulation environments. As shown in the figure, a low packet error rate can be achieved with high SNR or by adopting a lower MCS level. Note that in the network areas with densely populated stations and APs, interference among uplink transmissions may significantly lower the throughput performance. When multiple APs independently provide wireless connectivity services to their associated stations, each uplink transmission could work as an interferer to other uplink transmissions. Instead of non-cooperative uplink transmissions, we propose to utilize a cooperative uplink method that centrally manages uplink scheduling decisions.

Figure 4 illustrates the average network throughput performance per time slot concerning the number of stations attempting to transmit uplink data. In the simulation setup, 20 APs are deployed in the network to offer uplink connectivity services to stations. Consequently, the simulation outcomes reveal different trends for scenarios with fewer than 20 stations and those with more than 20 stations. When the number of stations is less than 20, the random cooperative method shows better performance than the proposed Lyapunov drift-plus-penalty-based method on average. This is attributed to the random method’s ability to select the solution with the highest expected throughput from a randomly chosen feasible set of solutions, whereas the proposed method focuses on optimizing Lyapunov drift, which accounts for transmission stability based on queue status and anticipated uplink success probability. Furthermore, the proposed method considers the required cooperation delay, leading to decreased throughput performance compared to the random method. In non-congested network areas, where stations and APs are sparsely deployed for wireless connectivity services, the constraints involved in uplink cooperation, such as cooperation delay and stability, may impose strict limitations on viable solutions. This can lead to a reduction in throughput performance. The rationale behind this is that in such environments, the influence of interfering signals from nearby areas does not substantially compromise throughput performance. However, as the network experiences increased congestion due to a higher number of stations, the performance of the random method deteriorates, while the proposed method demonstrates stable or improved performance. Specifically, the proposed method with a trade-off parameter of V=0.1 exhibits superior throughput compared to other methods when the number of stations is 18 or more in the simulated environments. For scenarios with over 24 stations, the proposed method with a trade-off parameter of V=0.3 shows better throughput performance than the random method. Note that decreasing *V* places more emphasis on stability represented by Lyapunov drift, while increasing *V* prioritizes the required cooperation delay at the fronthaul network in scheduling decisions as represented in (Equation 12). The results highlight that the proposed cooperative scheduling method with well-adjusted trade-off parameter *V* enhances network throughput performance in dense networks. Conversely, the non-cooperative scheduling method consistently exhibits the lowest throughput performance across different numbers of stations. This implies the potential of cooperation to enhance network performance in densely populated areas with numerous APs and stations.

Figure 5 depicts the cooperation delay of the proposed method with different trade-off parameters and the random cooperative method in relation to the number of stations. The simulation results for the non-cooperative method are not included as this method does not entail additional delays for cooperation. As previously mentioned, an increase in the trade-off parameter *V* results in lower delay because the proposed method prioritizes delay more significantly during the scheduling process. Consequently, the proposed method with V=0.1 exhibits a longer delay compared to the methods with V=0.3 and V=0.5. On the other hand, across all three cases with varying trade-off parameters, the proposed method demonstrates a shorter cooperation delay than the random cooperative method. In the simulated environments, the delay follows the gamma distribution with a shape parameter of $g_{sh}=4$ and a scale parameter of $g_{sc}=2$, resulting in a mean delay value of 8. The observed cooperation delay of the random method displaying values close to 8 aligns with expectations, as the random distribution selects solutions without taking into account delay considerations. The results also indicate that if the fronthaul network connecting the cooperative scheduler and the cooperative APs experiences congestion from heavy traffic or expands with additional switches, cooperation could encounter challenges. Therefore, it is essential to consider the required delay for information exchange when implementing cooperative scheduling in dense networks.

Figure 6 illustrates the trade-off relationships of the proposed Lyapunov drift-plus-penalty-based cooperative method with varying trade-off parameters, *V*, and the random cooperative method when the number of stations, S, is equal to or greater than 20 in the simulated scenarios, i.e., |S|∈{20,22,24,26,28,30}. The results demonstrate that the proposed method with V=0.3 exhibits comparable throughput performance to the random cooperative method, achieving throughput levels ranging from 200 to 300 bytes per time slot. However, there are significant differences in the required cooperation delay to achieve these throughput levels. While the proposed method necessitates delay values of around 4, the random cooperative method averages delay values of 8. On the other hand, the proposed method with a trade-off parameter of V=0.1 requires delay values of around 4 to achieve even higher throughput performance exceeding 300 bytes per time slot. This analysis, based on the trade-off considerations, highlights that the proposed Lyapunov drift-plus-penalty-based cooperative uplink scheduling effectively enhances network performance.

## 6. Conclusions

In this manuscript, we present a cooperative uplink scheduling method utilizing the Lyapunov drift-plus-penalty framework. In environments densely populated with independently functioning APs and stations, the independent transmission activities of each AP-station pair can cause substantial interference with other concurrent transmissions. To address this issue, a cooperative scheduler or a central coordinating entity can be implemented to control uplink scheduling decisions, thereby promoting more effective network management. To enhance the uplink transmission capability of the network, a specific subset of stations may be chosen, with several proximate APs allocated to support the uplink transmissions from a singular station. While cooperative scheduling has the potential to augment network throughput, it necessitates additional delays due to the requisite coordination. The Lyapunov drift-plus-penalty-based approach we propose takes into account both the anticipated probabilities of successful uplink transmissions and the associated cooperation delays within its scheduling considerations, thereby balancing these trade-offs. The IEEE 802.11ax standard grounded simulations demonstrate that the proposed method substantially enhances network performance by effectively balancing throughput gains and delay costs. In future research, we intend to apply the insights obtained from this study to the development of learning-based scheduling algorithms for cooperative or cloud Wi-Fi networks. The Lyapunov-drift-plus-penalty-based scheduling from this research could serve as the basis for designing a reward function in reinforcement learning models. Alternatively, the aspects of transmission stability and cooperation delays could be features for a deep learning-based scheduling decision. To evaluate the practical application of such models, we plan to conduct real-world experiments using a test bed configured with software-defined radio hardware and a cloud server for cooperative scheduler implementation. We aim to evaluate the performance of scheduling decisions influenced by the Lyapunov drift-plus-penalty framework, particularly in congested environments with a high density of stations and APs.

## Figures and Tables

**Figure 1 sensors-24-02399-f001:**
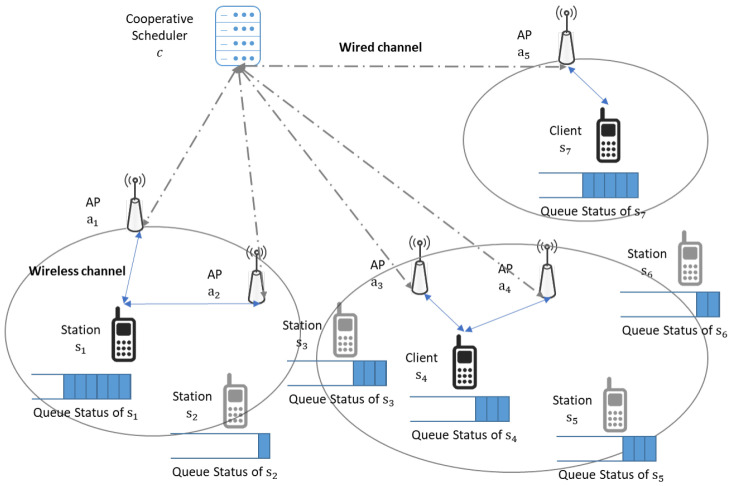
Uplink transmissions through both wireless and wired channels in centralized Wi-Fi networks.

**Figure 2 sensors-24-02399-f002:**
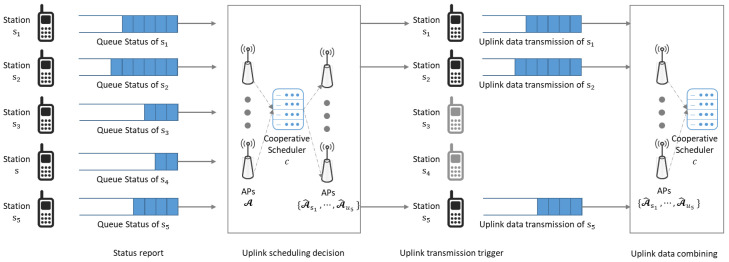
The procedure of cooperative uplink transmissions in cooperative Wi-Fi networks.

**Figure 3 sensors-24-02399-f003:**
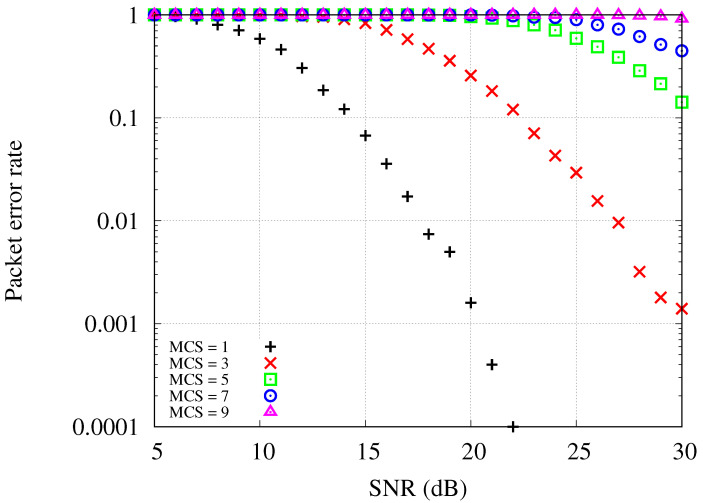
Packet error rate of IEEE 802.11ax in the constructed simulation environments.

**Figure 4 sensors-24-02399-f004:**
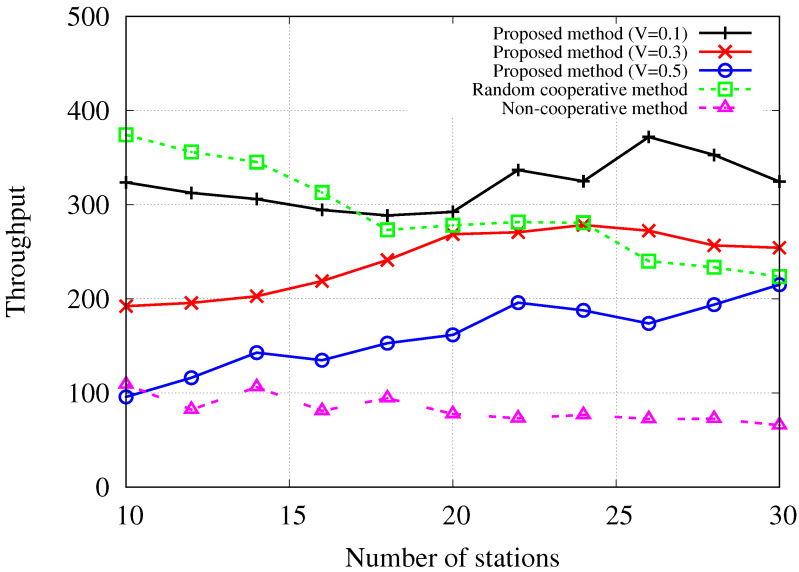
Throughput [bytes/timeslot] performance with regard to the number of stations when |A|=20.

**Figure 5 sensors-24-02399-f005:**
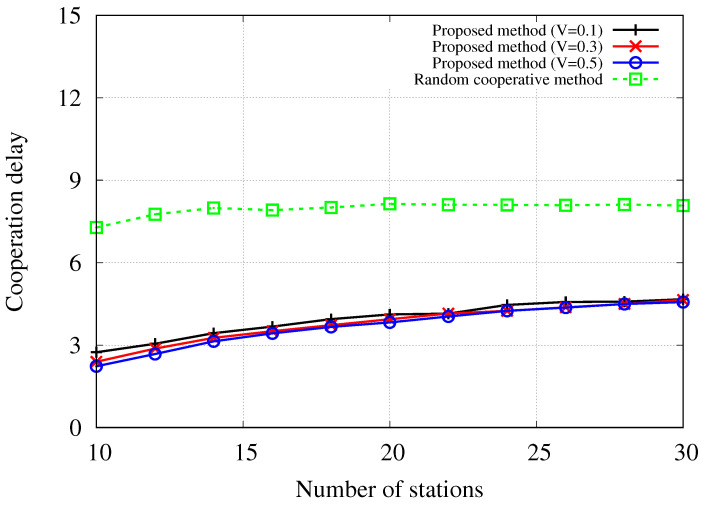
Required cooperation delay [μs] with regard to the number of stations when |A|=20.

**Figure 6 sensors-24-02399-f006:**
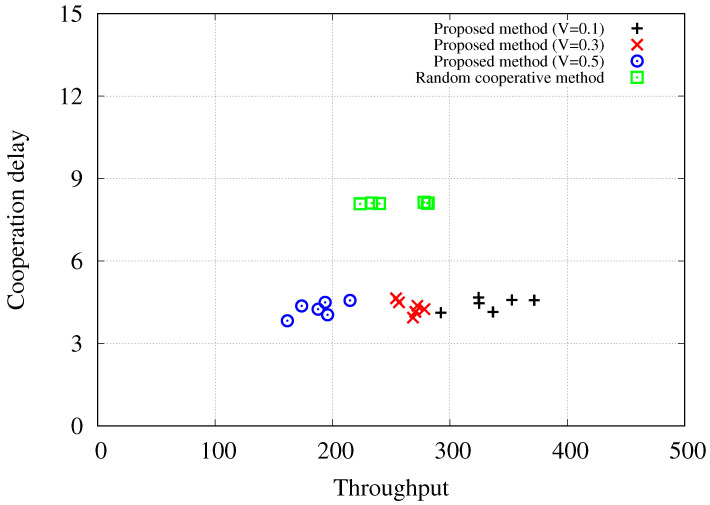
Trade-off relationship representation when |A|=20 and |S|≥20.

**Table 1 sensors-24-02399-t001:** IEEE 802.11ax-based simulation parameters.

Parameter	Value
Carrier frequency	5.25 [GHz]
Channel bandwidth	40 [MHz]
Channel coding	Low-density parity-check code (LDPC)
The number of tx/rx antennas	Tx antennas: 2, rx antennas: 2
Pathloss exponent	2 if transmit-receive node distance ≤ 5 m 3.5 if transmit-receive node distance > 5 m
Small-scale fading	Rayleigh fading
Transmission power	25 [dBm]
MCS	3
PSDU size	1024 [Bytes]

## Data Availability

Data are contained within the article.

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
