# Peer review of "Lyapunov Drift-Plus-Penalty-Based Cooperative Uplink Scheduling in Dense Wi-Fi Networks"

_sensors, 2024, doi:10.3390/s24082399_

Round 1

Reviewer 1 Report

Comments and Suggestions for Authors

The authors propose a Lyapunov drift-plus-penalty-based cooperative uplink scheduling method that considers uplink queue and scheduling delays for cooperative uplink transmission of multiple access points (APs) and stations in a high-density network environment. The proposed central coordination entity can manage the network to improve network performance, such as in the IEEE 802.11ax standard.

The paper is well-written and easy to understand for the reader. However, there are a few comments for quality improvement.

- C1. The authors propose a scheduling algorithm to maximize the uplink throughput considering queues, channels, and delays. However, Figure 2 seems to show that the station with the most data in the queue is selected first. Does the simulation result actually work as shown in Figure 2?

- C2. The authors argue in Figure 4 that the Random Cooperative Method outperforms the proposed methods because it selects stations that are expected to have higher throughput when the number of stations is less than the number of APs. However, the exact definition of the Random Cooperative Method is not described in the paper(Section 5).

- C3. Please add an abbreviation explanation for MCS = Modulation Coding Scheme level.

Author Response

First of all, we deeply appreciate all the efforts of the reviewer for invaluable comments, which have significantly improved the quality of our manuscript entitled “Lyapunov Drift-Plus-Penalty-Based Cooperative Uplink Scheduling in Dense Wi-Fi Networks”. We have tried our best to address all the issues raised by the reviewer.

Reviewer 2 Report

Comments and Suggestions for Authors

The authors proposed a Lyapunov drift-plus-penalty framework-based cooperative uplink scheduling method for densely populated Wi-Fi networks. Generally, the paper is well presented and easy to follow. The results seem interesting. However, I have the following comments:

1) The abstract should be entirely reformulated in a way to reduce the general context and give more details about the proposed system. The main findings of the paper should be mentioned as well.

2) The Introduction section should include more references.

3) In the related work section, the authors describe the existing techniques without discussing mentioning their limitations and the novelty of the proposed system compared to the state of the art.

4) In Algorithm 1, the randomness selection of the APs (line 5) could not ensure a convergence of the proposed model and having the optimal solution.

5) I miss a study about the noise impact to the proposed system.

6) The obtained results should be compared to the state-of-the-art techniques to show its relevance.  

Comments on the Quality of English Language

Minor editing of English language required

Author Response

(The authors gave the same response as above.)
